# Uncertainty in El Niño-like warming and California precipitation changes linked by the Interdecadal Pacific Oscillation

Lu Dong [1✉], L. Ruby Leung [1✉], Fengfei Song [1] & Jian Lu [1]

Marked uncertainty in California (CA) precipitation projections challenges their use in adaptation planning in the region already experiencing severe water stress. Under global warming, a westerly jet extension in the North Pacific analogous to the El Niño-like teleconnection has been suggested as a key mechanism for CA winter precipitation changes. However, this teleconnection has not been reconciled with the well-known El Niño-like warming response or the controversial role of internal variability in the precipitation uncertainty. Here we find that internal variability contributes > 70% and > 50% of uncertainty in the CA precipitation changes and the El Niño-like warming, respectively, based on analysis of 318 climate simulations from several multi-model and large ensembles. The Interdecadal Pacific Oscillation plays a key role in each contribution and in connecting the two via the westerly jet extension. This unifying understanding of the role of internal variability in CA precipitation provides critical guidance for reducing and communicating uncertainty to inform adaptation planning.

---

[1] Atmospheric Sciences and Global Change Division, Pacific Northwest National Laboratory, Richland, Washington, USA.
✉email: donglu.atmosphere@gmail.com; ruby.leung@pnnl.gov

Located on the western edge of the North America continent and influenced by the Pacific storm tracks, California (CA) has a distinct precipitation annual cycle, with a large fraction of precipitation falling within the winter season (December–January–February). Hence, winter precipitation is vital to the agriculture, ecosystems, and water resources in the region. Despite the very certain sign of precipitation frequency and intensity changes from multimodel ensembles, with decreased frequency and increased intensity, large uncertainty in the changes of annual or winter CA precipitation amount is evident due to the superposition of these two opposite contributions[1–4]. Influenced by both tropical forcing and mid-latitude westerlies[5], the intermodel spreads of both the signs and magnitudes of CA precipitation changes under warming are large[2,3,6–13].

Climate model projection uncertainty has been robustly partitioned into its different sources at global scale[14,15], but such partitioning may be highly variable at local-to-regional scale. Understanding the fractional contribution of uncertainty from different sources is important for informing the use of climate projections for regional adaptation planning. The total uncertainty conflates contributions of uncertainty from internal variability, the model response to forcing, and the emission scenarios[14–17]. With climate change projections conditioned on the emission scenarios, which are developed based on storylines of socioeconomic changes with no probabilities assigned, progress can be made in understanding uncertainty by focusing on the internal variability and model response uncertainty. We emphasize the decomposition of uncertainty into components of externally forced vs. internal variability because uncertainty in the response to external forcing is an important and potentially reducible uncertainty factor for targeted future research through model development and observational constraints. Although uncertainty from internal variability is irreducible, improving the decadal prediction of the relevant internal modes may also potentially reduce uncertainty in predicting the decadal trends in CA precipitation in the near future.

Separating the uncertainties caused by the model response and internal variability is difficult in traditional multimodel ensembles from the Coupled Model Intercomparison Project phase 5 (CMIP5; ref. [18]) and phase 6 (CMIP6; ref. [19]), as most models only include a small number of realizations, which can not faithfully represent the range of internal variability[20]. While selecting subsets of the CMIP models may reduce model uncertainty to provide more consistent projections of future CA precipitation[11,21], the internal variability is even more under-represented by the smaller subsets of models. The advent of large ensembles from several climate models presents an opportunity for isolating the internal variability from the model response uncertainty[22].

Internal variability, which arises from processes intrinsic to the atmosphere, the ocean, and the coupled ocean-atmosphere system via dynamic and thermodynamic interactions, makes an appreciable contribution to the precipitation changes on a decadal timescale[17,23], especially at smaller spatial scales[24]. The dominant effect of internal variability in the 2010–2015 CA megadrought has been broadly recognized[25,26], although anthropogenic warming is argued to enhance the probability of severe drought[27,28]. The Interdecadal Pacific Oscillation (IPO), or Pacific Decadal Oscillation (PDO), is a leading mode of internal variability at the decadal timescale featuring sea surface temperature (SST) variability in the Pacific Ocean. The IPO can be viewed as a manifestation of the integrated influences of the Pacific Ocean, including the El Niño-Southern Oscillation (ENSO)[29,30]. As described in previous studies[31–33], the IPO impact on CA precipitation is manifested via the interdecadal

modulation of ENSO teleconnections. Based on 21 CMIP3 models, a study[17] suggested that more than half of the intermodel spread in the precipitation changes under global warming over most of the extratropical regions are contributed by internal variability, which is estimated based on a single set of large ensemble simulations. In contrast, based on 36 CMIP5 models, another study[34] concluded that internal variability does not contribute substantially to the intermodel spread over broad regions, including CA. Similar conclusions were obtained by comparing the intermodel range from CMIP5 models with the intermember range from a single set of large ensemble simulations[11]. These controversial results motivate a need to combine the CMIP5 and CMIP6 models with several large ensemble simulations to quantify the contribution of internal variability to the total uncertainty of CA precipitation changes.

Physically, future changes in CA winter precipitation under warming are related to an eastward extension of the North Pacific westerly jet that steers more storms towards CA, analogous to the El Niño-like teleconnection[7,11,34]. However, previous studies[7,11,12,34] based on CMIP5 models could not establish a close intermodel relationship between the CA precipitation changes and the El Niño-like warming (i.e., stronger SST warming in the tropical eastern Pacific relative to the western Pacific) under global warming, and suggested that the relationship may have been obscured by model deficiencies, such as the CA precipitation sensitivity to Niño 3.4 SSTs, CA precipitation climatology, and possible overestimation of tropical convection[11,35]. Similar to CA winter precipitation, there is also substantial uncertainty in the El Niño- vs. La Niña-like (i.e., stronger SST warming in the tropical western Pacific relative to the eastern Pacific) warming pattern over the tropical Pacific under global warming[36–38]. In contrast with the controversial role of internal variability in the CA winter precipitation, the possible role of internal variability in the future El Niño-like warming pattern has yet to be brought to the fore, although there is an ongoing debate on the relative roles of model bias[38,39] vs. internal variability[40–42] in the La Niña-like warming pattern observed in the recent decades. With the convolved contributions of model uncertainty and internal variability to the large uncertainty in the El Niño-like warming and CA precipitation, CMIP simulations offer limited opportunities to isolate the roles of internal variability in their respective future changes and their relationships. We conjecture that quantifying and understanding the contribution of internal variability to uncertainty in the El Niño-like warming may hold a key to better understanding and reducing the uncertainty of CA precipitation changes in the future.

In this work, a total of 318 simulations including the large ensembles and multimodel ensembles show a marked contribution of internal variability of >80% and >70% to the total uncertainty in the decadal trends and future changes of CA precipitation, respectively. Importantly, internal variability also contributes >50% to the total uncertainty in the future change of El Niño-like warming pattern. Among the internal variability, the IPO is key to connecting the uncertainties of CA precipitation and El Niño-like warming pattern through its modulation of the Aleutian low and westerly jet extension over the North Pacific.

## Results

To test our conjecture, we use the large ensemble simulations from three climate models (CESM1, CanESM2, MPI-ESM) with a total of 190 members, comparable to the total of 128 members in the CMIP5 and CMIP6 multimodel ensembles (See Methods). The three climate models of the large ensembles perform very well in capturing the climatological CA precipitation and westerly jet stream (Supplementary Fig. 1). Note that the large ENSO bias found in CCSM3 has been significantly reduced in CCSM4,

CESM1, and CESM2, and the spatial patterns of the IPO, as well as their relationships to ENSO modulations are well simulated in the CESM models[43]. We find that the spatial patterns of the IPO simulated by all the three models show high pattern correlation coefficients (~0.7, statistically significant at the 99% level of confidence) with the observation over the Pacific Ocean. The interdecadal variability of the IPO is also reasonably well simulated as indicated by comparing the power spectra between the three models (grey lines) and observation (black lines in Supplementary Fig. 2). Furthermore, teleconnections of the IPO can be realistically represented in most climate models[44]. We estimate the total uncertainty by the spread of all 128 members from CMIP5 and CMIP6 (Supplementary Table 1) following the previous studies[7,11,34], while internal variability is estimated by the intermember spread of the large ensemble from each of the three models (see Methods).

**Internal variability dominates uncertainty in CA winter precipitation decadal trends.** Over the northern mid-latitudes, the historical precipitation trend over the U.S. west coast during 1979–2019 is subject to large uncertainty compared to other land regions, as indicated by the large standard deviation (STD) across all 128 members of the CMIP5 and CMIP6 models (Fig. 1a). Uncertainty from internal variability is comparable to uncertainty from the CMIP models not only over the U.S. west coast but also

in the North Pacific storm track region (Fig. 1b), which has been suggested to be closely linked to CA precipitation[7,34], underscoring the large contribution of internal variability to the total uncertainty. Precipitation trends in the near-future (2020–2060) and far-future (2061–2099) show similar spatial patterns (not shown). Focusing on CA, uncertainties in the decadal precipitation trends (Supplementary Fig. 3d–f) are larger than the mean trends (Supplementary Fig. 3a–c) in the past (1979–2019), near-future (2020–2060) and far-future (2061–2099) decades based on both the single-model large ensembles and the multimodel ensembles, indicating large uncertainty in CA precipitation decadal trends. The largest uncertainty occurs in winter, when precipitation peaks during the year. The total uncertainty in the CMIP models contains contributions from model response uncertainty (under a given emission scenario) and internal variability. As a first step in separating the two contributions, we compared the ratio of the STD from internal variability estimated based on the three single-model large ensembles relative to the total uncertainty estimated based on the CMIP models. Averaged across the three large ensembles, internal variability explains >80% of the total uncertainty in CA winter precipitation decadal trends, which is robust for the trends in the past, near-future, and far-future decades (Fig. 1c). The three large ensembles behave remarkably similarly (Supplementary Fig. 3g-i), demonstrating the importance of internal variability relative to the model response uncertainty in CA winter precipitation change on decadal timescales.

**Major contributions from the IPO and its mechanistic connection to CA precipitation.** To identify the internal climate mode primarily responsible for the uncertainty in CA winter precipitation change, we calculate the intermember regression of surface temperature trend onto the CA precipitation trend based on the three large ensembles. The regression features warming in tropical central-eastern Pacific and cooling in North Pacific (Fig. 2a), which resembles the positive IPO or PDO[31] pattern (Supplementary Fig. 4a). Note that the highest correlation occurs north of the Bering Sea, which is unlikely to be caused by the IPO, as one previous study[45] found that surface temperature response is absent over there when the observed tropical SST featuring a negative IPO is prescribed in an atmospheric model. The dipole pattern is consistent among the three large ensembles and across the past, near-future, and far-future decades (Supplementary Fig. 5). Therefore, the IPO decadal trend may contribute importantly to uncertainty in CA winter precipitation trend. Although the influences of both the Aleutian low at the surface[31,46] and the upper-tropospheric westerly jet stream over the North Pacific[7,11,34] on the CA winter precipitation are well established, it is still unclear to what extent they contribute to the intermodel spread of the CA winter precipitation trend through the IPO decadal variation. Hence we further explore the mechanism for how the IPO modulates the uncertainty in decadal precipitation change over CA. Overall, simulations with a larger positive IPO trend is correlated with a larger CA precipitation increase (Fig. 2b) via atmospheric teleconnection by deepening the Aleutian low (Fig. 2c) and instigating an eastward extension of the westerly jet stream over the North Pacific (Fig. 2d). Based on the 50 CanESM2 members during 1979–2019, the intermember relationship shows the linear trend in CA precipitation is negatively correlated with that in the Aleutian low (r = −0.77, black in Fig. 2e) and positively correlated with that in the westerly jet extension over the North Pacific (r = 0.76, red in Fig. 2e). Consistently, linear trends in the IPO show a significant relationship with those in the Aleutian low (r = −0.68, black in Fig. 2f) and westerly jet extension (r = 0.79, red in Fig. 2f),

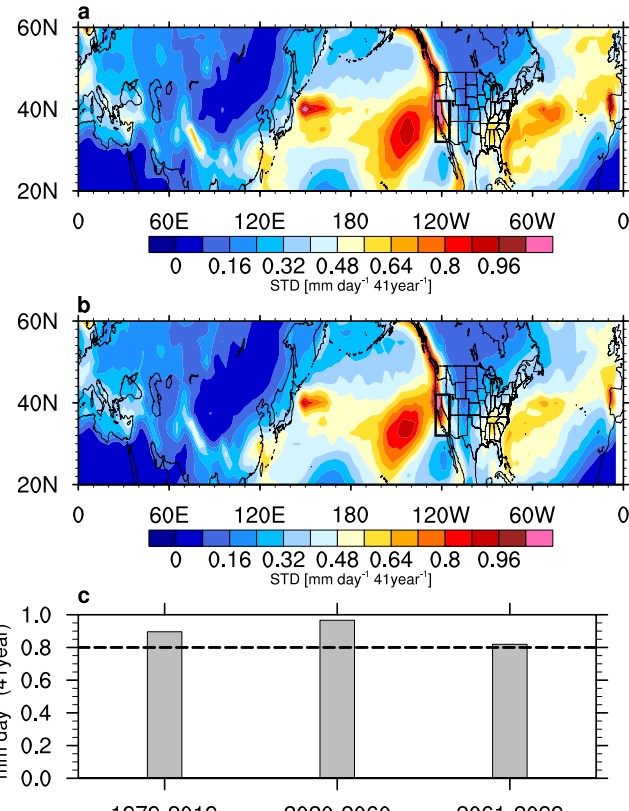

**Fig. 1 Effect of internal variability on the large uncertainty of winter precipitation decadal trends over California. a** Total uncertainty of winter precipitation trend during 1979–2019 based on the 128 members of CMIP5 and CMIP6 (CMIPs). **b** Internal uncertainty based on the average of intermember standard deviation (STD) from three large ensembles (including 100 MPI-ESM, 40 CESM1, and 50 CanESM2 simulations). **c** Fraction of the total uncertainty in California winter precipitation trend during 1979–2019, 2020–2060, 2061–2099 explained by internal variability based on the average of three large ensembles. Units: mm day$^{-1}$ 41year$^{-1}$.

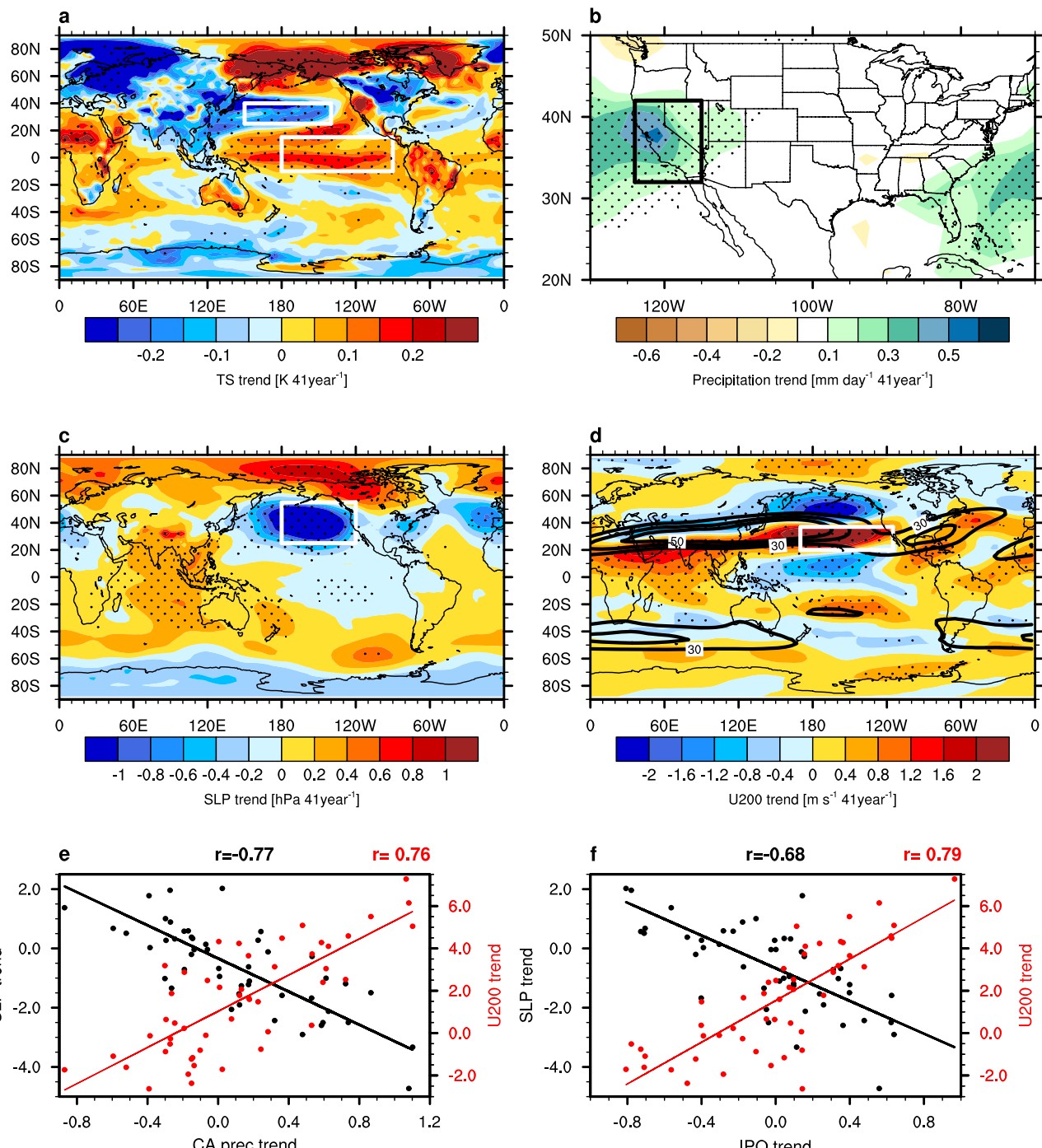

**Fig. 2 The dominant internal mode influencing the uncertainty of California precipitation decadal trends and the mechanism.** Intermember regressions of **a** surface temperature (TS) trend (K 41year$^{-1}$) onto the California averaged precipitation (CA prec) trend. Intermember regressions of the linear trend in **b** precipitation (mm day$^{-1}$ 41year$^{-1}$), **c** sea level pressure (SLP; hPa 41year$^{-1}$), **d** 200hPa zonal wind (U200; m s$^{-1}$ 41year$^{-1}$) onto the Interdecadal Pacific Oscillation (IPO) trend. Scatterplots of **e** the California precipitation trend (x-axis) and **f** the IPO trend (x-axis) versus the SLP trend over the Aleutian low (black, left y-axis) and the zonal wind at 200 hPa (U200) trend over the jet extension region (red, right y-axis). The climatological jet stream based on the historical simulations is shown in black contours in panel **d**. All the trends are based on the 50 members of CanESM2 during 1979–2019. The regression lines and the intermember correlations (r) are shown in corresponding color on top of the bottom panels. The averaging areas for calculating the quantities shown in the bottom panels are indicated as rectangles in panels **a** to **d**.

reconciling with the atmospheric circulation that modulates the decadal trends in CA precipitation under internal variability. These intermember relationships largely hold for all three large ensembles as well as the three periods (Supplementary Figs 6, 7). In particular, correlation coefficients between the trends of Aleutian low and the IPO trends for 1979–2019 are −0.74

(CESM1), −0.68 (CanESM2), and −0.63 (MPI-ESM) (Supplementary Fig. 7a–c), all statistically significant at the 99% level of confidence. They suggest that the atmospheric circulation mechanism plays a dominant role in the uncertainty of decadal precipitation trends under internal variability. Thus, the IPO is the key internal mode influencing the uncertainty in the decadal

trend of CA winter precipitation in the past, near-future, and far-future.

To compare the relationship between the IPO and CA winter precipitation directly, the intermember correlation of the 41-year trends for the past, near-future, and far-future decades is examined based on the three large ensembles (Supplementary Fig. 8). Positive correlations are found for all the three large ensembles during the three periods, with correlation coefficients of 0.34, 0.57, and 0.44 for CESM1, CanESM2, and MPI-ESM, respectively, during 1979–2019, all statistically significant at the 95% confidence level. Recognizing the mechanistic and statistically significant connections between the IPO trends and the CA precipitation decadal trends, can the IPO be used to constrain the uncertainty of CA precipitation change under warming?

**Constraining the CA precipitation decadal trends by the IPO reduces uncertainty.** In previous studies, the model ability to capture the ENSO-precipitation relationship at inter-annual timescale is used as an emergent constraint to reduce the uncertainty in precipitation projections from the model bias perspective[11,47]. Here, we attempt to constrain the CA precipitation decadal change by considering the role of the IPO. Different from emergent constraints that use observations to constrain uncertainty in future projections due to model biases, the state of the IPO can be used to constrain uncertainty in future projections related to internal variability. To quantify the uncertainty in CA precipitation trends explained by the IPO, we exclude the IPO's influence by removing the CA precipitation variations that are linearly related to the IPO index in each realization of the large ensembles (See Methods). The histogram and the fitted frequency distribution of CA precipitation trends narrow noticeably after removing the IPO's influence, with the STD reduced by 0.3%, 16%, 12% for 1979–2019, 26%, 11%, 16% for 2020–2060, and 12%, 25%, 10% for 2061–2099, based on the CESM1, CanESM2, MPI-ESM ensembles, respectively (black and red in Fig. 3a–c, Supplementary Fig. 9, Supplementary Table 2). Although these reductions of STD are modest, as only the variability that is linearly related to the IPO is removed, they are non-negligible and indicate the role of the IPO in increasing the chance of both the extreme positive and extreme negative precipitation trends. Conditioning on the observed IPO trend of $-1.0$ K $(41\text{year})^{-1}$ during 1979–2019 (Supplementary Fig. 4b), the distribution of CA precipitation trends shifts towards drying (blue line in Fig. 3a and Supplementary Fig. 9a, b), with the mean (blue dot) falling between the observed drying trends based on two observation datasets (purple dots in Fig. 3a and Supplementary Fig. 9a, b). This implies a dominant role of the IPO in the observed decadal drying trend and cautions the interpretation of model-observation differences as model biases, as internal variability may account for a significant fraction of that difference. To further support this statement, members from the three large ensembles that produce drying trends in CA no less than the observed drying of $-0.44$ mm day$^{-1}$ 41year$^{-1}$ estimated by GPCP are used to composite their SST trends (Supplementary Fig. 10). Averaging all the members which represents the response to the external forcing, the SST trend during 1979–2019 features warming in most of the global ocean based on all the three large ensembles (Supplementary Fig. 10a–c). In contrast, the SST trend with external forcing removed in the members that reproduce the observed drying in CA features a negative IPO pattern in all the three models, similar to the observed IPO during 1979–2019 (Supplementary Fig. 10d–f), confirming the critical role of the IPO in the recent CA drying. The latter suggests that uncertainty in projecting CA precipitation change in near-future could be reduced with improved decadal prediction of the IPO.

The contributions of external forcing and the positive-to-negative phase transition of the IPO in the drying CA trend during 1979–2019 are further assessed quantitatively. Observed precipitation in CA shows a significant drying trend of $-0.61$ mm day$^{-1}$ $(41\text{year})^{-1}$, reducing the mean precipitation during the 41-year period by ~28% of the climatological precipitation, based on the average of GPCP[48] and CMAP[49] (Fig. 3d). Neither the multi-models (green bar) nor any single-model large ensembles (grey bars) under external forcing can reproduce the observed drying trend (Fig. 3d). Taking the observed IPO transition during 1979–2019 into account (See Methods), all three large ensembles can well reproduce the observed drying trend in CA precipitation based on the multi-member average, with magnitudes comparable to the observation (orange bars in Fig. 3d). Therefore, the drying over CA in the past decades is dictated by the positive-to-negative IPO phase transition, which overshadows the insignificant effect of external forcing (Fig. 3d). However, with continued warming in the future, external forcing has stronger and more significant effect on the decadal trend of CA winter precipitation (Fig. 3e), which may overwhelm the effect of the IPO.

As uncertainties contain contributions from model response uncertainty and internal variability[14,15,17], it is important to compare their time-dependent relative contributions. Comparing the time evolution of these uncertainties, the total uncertainty of CA precipitation decadal trends increases gradually with warming, while the internal variability contribution, especially the IPO component, remains stable (Fig. 3f). Internal variability dominates the total uncertainty of the CMIP simulations, indicating a more important role of internal variability than model response uncertainty. However, the time series of the total uncertainties estimated based on the CMIP models occasionally fall below the time series of the internal variability (grey solid line vs. black dashed line in Fig. 3f). This suggests that the total uncertainty is likely underestimated by using the CMIP models due to the limited number of simulations from each model[20], but our analysis suggests that most of the intermodel spread in the CA precipitation from the CMIP models can be represented by internal variability. Importantly, the IPO-related uncertainty explains ~50% of the internal uncertainty, accentuating the dominant contribution of the IPO to the internal uncertainty.

**Role of the IPO in the future changes of CA precipitation and El Niño-like warming pattern.** Having demonstrated the significant role of the IPO in the large internal variability uncertainty in CA winter precipitation decadal trends, we further quantify the role of internal variability in the uncertainty of future changes of CA winter precipitation, which have been more broadly investigated in previous studies[7,11–13,34,50]. Here the differences between 2085–2099 from the RCP8.5/SSP585 simulations and 1986–2000 from the historical simulations are used to represent the future changes under global warming. Consistent with the 41-year linear trend, internal variability accounts for a marked contribution (>70%) to the total uncertainty in the future change of CA precipitation based on three large ensembles as well as their mean (Fig. 4a). The El Niño-like SST warming pattern has been suggested to contribute to the uncertainty of the westerly jet extension related with CA precipitation change[12]. Although the intermodel relationship between the El Niño-like warming and CA precipitation change is not significant among the CMIP5 models (Supplementary Fig. 11a), consistent with previous studies[7,11,34], all three large ensembles show a statistically significant intermember relationship between the two variables (Fig. 4b, Supplementary Fig. 11c–e) at the 99% confidence level. The latter suggests an important contribution of the El Niño-like warming pattern to the internally-induced uncertainty of CA

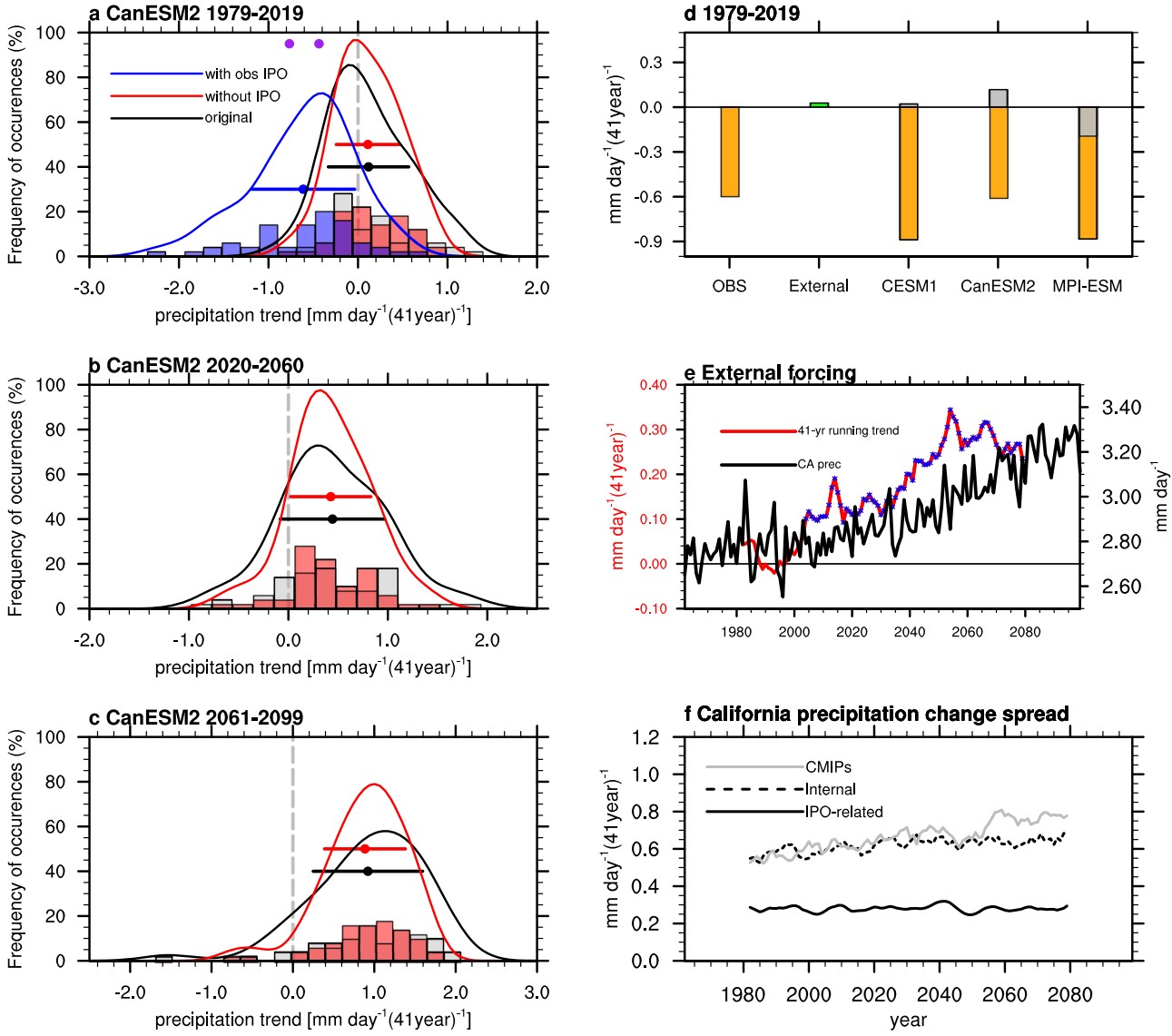

**Fig. 3 Constraining the California precipitation decadal trends by the Interdecadal Pacific Oscillation (IPO). a–c** Histograms (bars) and 100-bin fitted frequency distribution (lines) of the California winter precipitation trends during **a** 1979–2019, **b** 2020–2060, and **c** 2061–2099 based on 50 members of CanESM2. The gray bars and the black fitted curves show the frequency of occurrence of the original trends; the red bars and curves are the same but with the IPO's influence removed through linear regression against the IPO index in the individual runs; the blue bars and curves are the same but including the observed IPO trend for 1979–2019. The dots and error bars denote the ensemble mean and one standard deviation of the distribution represented by the corresponding color. The purple dots denote the observed precipitation trend based on GPCP ($-0.44$ mm day$^{-1}$ 41year$^{-1}$) and CMAP ($-0.77$ mm day$^{-1}$ 41year$^{-1}$) datasets. **d** Linear trend of California winter precipitation during 1979–2019. Observation (OBS) is the average of GPCP and CMAP; External is the average of CMIP5, CMIP6, and the three large ensembles (green bar). Trends for the three large ensembles account for the observed IPO trend (orange bars), with the gray bars showing the response to external forcing based on each large ensemble (multi-member mean of each large ensemble). **e** Time series of California precipitation (CA prec; in mm day$^{-1}$, black, right y-axis) and its 41-year running trend (mm day$^{-1}$ 41year$^{-1}$, red, left y-axis), with the significant trend at the 95% level of confidence shown in blue dots under external forcing. **f** Time series of total uncertainty estimated based on the CMIP5 and CMIP6 models (solid grey line), internal variability (dashed black line), and IPO-related internal variability (solid black line) of the 41-year running trend of California precipitation.

precipitation change by modulating the westerly jet extension (Supplementary Fig. 11f–h). This contribution of El Niño-like warming pattern to uncertainty in CA precipitation change may have been masked by the large model response uncertainty in the CMIP5 models. Notably, the CMIP6 models show a more significant intermodel relationship ($r = 0.59$, Supplementary Fig. 11b) compared to the CMIP5 models ($r = 0.23$). Whether model improvement or reduced model response uncertainty from CMIP5 to CMIP6 has contributed to the change in the relationship needs further investigation in the future.

Most models project an El Niño-like warming pattern under global warming, except for MPI-ESM which projects a La Niña-like warming pattern (Fig. 4c) consistent with its projection of a drying trend under external foring (grey bar in Fig. 3d). It is noteworthy that the CMIP6 models project a stronger El Niño-like warming than the CMIP5 models based on the multimodel mean, but with a large intermodel spread in both. Even for a single model, the intermember spread is also comparable to the mean, underscoring the large uncertainty from internal variability in the future change of the SST warming pattern. The fractions of the total

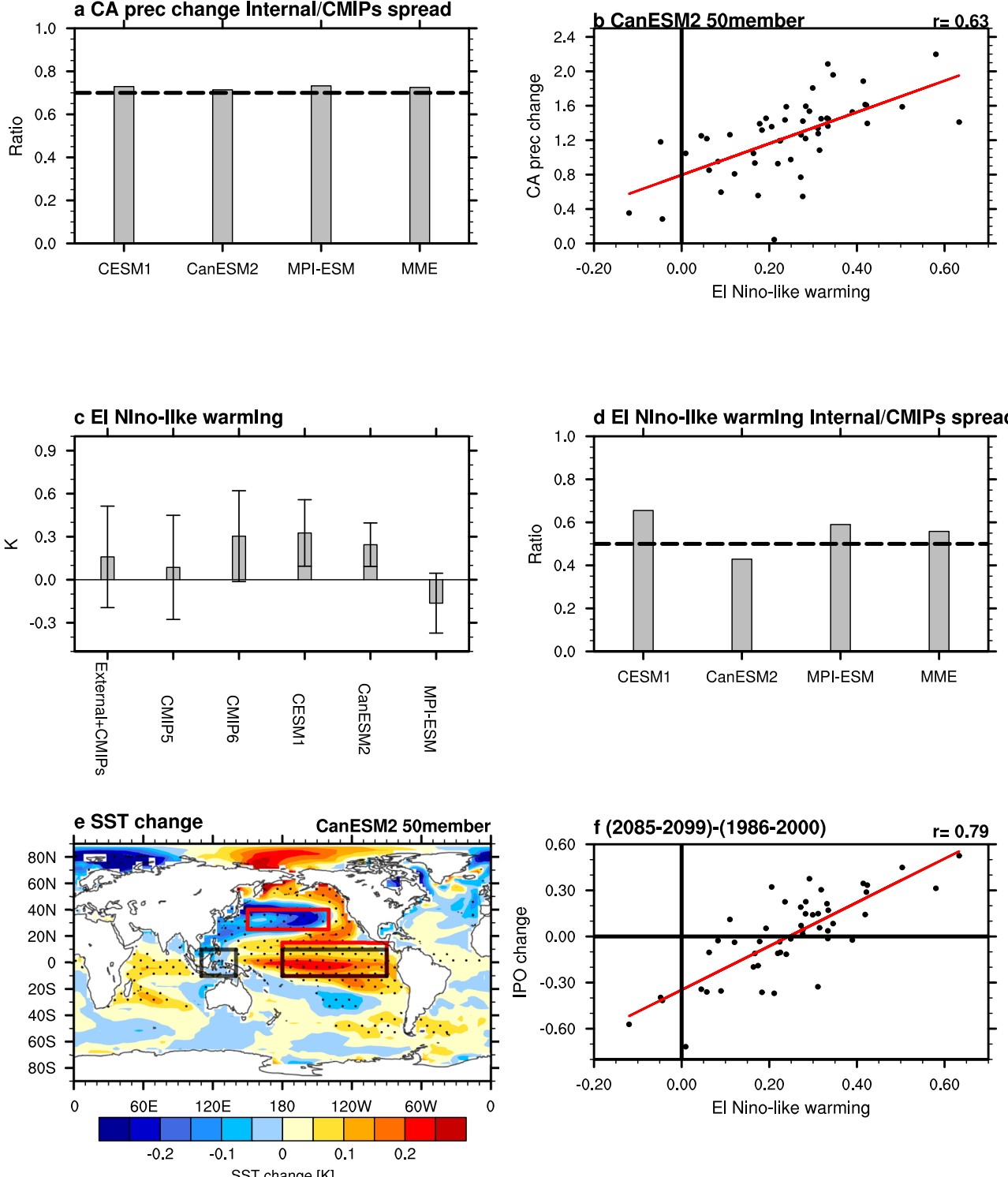

**Fig. 4 Role of the Interdecadal Pacific Oscillation (IPO) in the uncertainty of the El Niño-like warming pattern in the future. a** Fraction of the total uncertainty in California precipitation (CA prec) change explained by internal variability based on three large ensembles and their mean. **b** Scatterplots of the El Niño-like warming (K, x-axis) versus California precipitation change (mm day$^{-1}$, y-axis) based on 50 members from CanESM2. **c** The El Niño-like warming pattern change (K) based on all members, 57 CMIP5 members, 71 CMIP6 members, 40 CESM1, 50 CanESM2 and 100 MPI-ESM, respectively. Grey bars denote the ensemble mean, and error bars denote one standard deviation. The first term denotes the effect of external forcing (average of five large ensembles, grey bar) and the total uncertainty of 128 CMIP5 and CMIP6 simulations (error bar). **d** Fraction of the total uncertainty in El Niño-like warming explained by internal variability based on three large ensembles and their mean. **e** Intermember regressions of sea surface temperature (SST) change (K) onto the El Niño-like pattern change based on 50 members from CanESM2. **f** Scatterplots of the El Niño-like warming (K, x-axis) versus the IPO change (K, y-axis) based on 50 members from CanESM2. The regression line is shown as the red line and the intermember correlation (r) is shown at the top-right of panel **b** and **f**. The El Niño-like pattern is based on the SST inside the black rectangles, and the IPO is based on the SST inside the red rectangles in **e**. All the changes are based on the difference between RCP8.5/SSP585 (2085–2099) and historical (1986–2000).

uncertainty in the El Niño-like warming explained by internal variability are estimated based on the three large ensembles as well as their mean (Fig. 4d). Internal variability can explain >50% of the total uncertainty in the El Niño-like warming pattern in the future based on the three large ensembles mean.

We hypothesize that among the internal variability, the IPO may contribute substantially to the uncertainty of El Niño-like warming pattern in the future, which induces uncertainty in the CA precipitation change through atmospheric teleconnection. To test this hypothesis, we investigate the intermember regression of SST change onto the El Niño-like warming index (Fig. 4e). The regressed SST exhibits a pattern remarkably similar to that of the positive IPO phase, indicating a close relationship between the IPO change and the El Niño-like pattern[29]. The correlation coefficient between the IPO change and the El Niño-like warming pattern is 0.79 based on the 50 members of CanESM2, statistically significant at the 99% confidence level (Fig. 4f). Hence the IPO explains ~62% of the internally-induced uncertainty of the El Niño-like warming pattern based on CanESM2. The other two large ensembles based on CESM1 and MPI-ESM show consistent results supporting our hypothesis (Supplementary Fig. 12). As an internal climate mode, the IPO changes are symmetric about zero, averaging to nearly no change (see Methods), while the El Niño-like warming pattern has components of internal variability and the model response to external forcing, with the latter tending to be positive in CESM1 and CanESM2 and negative in MPI-ESM (Fig. 4f, Supplementary Fig. 12c, d). These results confirm the important contribution of the IPO to the uncertainty of the future change in El Niño-like warming pattern.

## Discussion

Located in the path of the Pacific storm tracks and significantly influenced by storms, CA winter precipitation is highly variable[51]. Using a large set of climate simulations including two multimodel ensembles (CMIP5 and CMIP6) and three single-model large ensembles for a total of 318 simulations, this study links the uncertain CA winter precipitation decadal trends and future changes to the uncertain El Niño-like warming pattern through their respective connections to internal variability. Specifically, internal variability accounts for >80% of the CMIP intermodel spread of CA winter precipitation decadal trends in the past (1979–2019), near-future (2020–2060), and far-future (2061–2099). Moreover, internal variability is estimated to account for more than half of the total uncertainty in the projected El Niño-like warming pattern and contributes to >70% of the intermodel spread in CA precipitation change in the future. The uncertainties in CA precipitation changes and El Niño-like warming are physically linked by the IPO, which connects the two by modulating the Aleutian low and westerly jet extension. Accounting for the positive-to-negative phase transition of the IPO during 1979–2019 by linear regression, the simulated CA precipitation trends are comparable to the observed drying trends. In addition, simulations that can reproduce the observed recent CA drying feature the negative IPO pattern.

Recognizing and understanding the relative contributions of internal variability and model response to the total uncertainty in CA precipitation projections can help focus our efforts in addressing uncertainty and improve communication of the uncertainty to stakeholders of the climate information. Although reducing the uncertainty of model response to external forcing may only reduce the uncertainty in CA precipitation projections by <30% (due to >70% of the uncertainty from internal variability), we have identified uncertainty in the El Niño-like warming response to external forcing as an important and potentially reducible uncertainty factor for targeted future research. This is hinted by the increased correlation between the El Niño-like warming and the CA precipitation projections in CMIP6 relative to CMIP5, although more detailed analysis of the reasons behind the difference is needed.

Based on the above analysis, the teleconnection between the IPO-related jet extension, persistent blocking high-pressure and CA drought can be inferred. In terms of the uncertainty from internal variability, the strong and significant negative correlations between the linear trends of the jet extension and the presence of persistent high-pressure exist for all three large ensembles as well as the three periods (Supplementary Fig. 13), demonstrating this teleconnection. It indicates that the positive-to-negative phase transition of the IPO may contribute to CA drought by inducing westward retreat of the jet and persistence of high pressure, with the latter steering advected moisture away from CA. In particular, the drying trend of $-0.61$ mm day$^{-1}$ (41year)$^{-1}$ during 1979–2019 has reduced the CA precipitation by ~28% of its climatological mean. Our findings highlight the importance of internal variability, especially the positive-to-negative phase transition of the IPO, in this observed drying trend and caution interpreting the role of model errors in model-observation differences in the historical simulations. Although uncertainty from internal variability is irreducible, given the long timescale of the IPO, improving its decadal prediction may potentially reduce uncertainty in predicting the decadal trends in CA precipitation in the near future and support stakeholders in planning for changing likelihood of extreme events such as flood and drought. Near-term predictions of the IPO based on initialized multimodel ensemble decadal hindcasts have shown some promises, with future improvements possible through community activities[52,53].

Lastly, our finding of the dominant role of internal variability in CA precipitation trends and projections is partly conditioned on the estimation of the total uncertainty based on the CMIP multimodel ensembles that reflect uncertainty from both model response and internal variability. Although this approach is also commonly used in many previous studies[7,11,34], comparison between the total uncertainty estimated using CMIP simulations and the internal variability estimated using the large ensemble simulations suggests that the total uncertainty is likely underestimated, due to the limited number of simulations from each model[20]. This calls for the need of large ensemble simulations from more modeling centers in the future to better quantify both model uncertainty and internal variability.

## Methods

**Models and datasets.** In this study, we use the monthly gridded precipitation data from the Global Precipitation Climatology Project Version 2.3 (GPCP, ref. [48]) and the CPC Merged Analysis of Precipitation Version 2002 (CMAP, ref. [49]), both covering the period of 1979–2019 with a horizontal resolution of 2.5° × 2.5°. Observed monthly SST data are taken from the Hadley Centre Sea Ice and Sea Surface Temperature dataset (HadISST), covering the period 1870 to 2019 with a horizontal resolution of 1.0° × 1.0° (ref. [54]).

To estimate the role of internal variability and model response uncertainty in the total uncertainty, we use monthly outputs from five simulation ensembles combining both the historical and future scenarios:

(1) Historical (1962–2005) and RCP8.5 (2006–2099) simulations of 57 members from 37 models from CMIP5 (Supplementary Table 1; ref. [18]).
(2) Historical (1962–2015) and SSP585 (2016–2099) simulations of 71 members of 37 models from CMIP6 (Supplementary Table 1; ref. [19]).
(3) Historical (1962–2005) and RCP8.5 (2006–2099) simulations of 50 members from the CanESM2 large ensemble project (ref. [55,56]).
(4) Historical (1962–2005) and RCP8.5 (2006–2099) simulations of 40 members from the CESM1 large ensemble project (ref. [57]);
(5) Historical (1962–2005) and RCP8.5 (2006–2099) simulations of 100 members from the MPI-ESM Grand Ensemble project (ref. [58]).

All the outputs from the CMIP5 and CMIP6 models are interpolated to a common 73 × 144 global grid using bilinear interpolation. The three sets of large ensemble simulations are the only ones providing 40 or more members, more than

the ensemble simulations from other models. The total number of members from the five simulation ensembles is 318.

**Estimating total uncertainty, internal uncertainty, and external forcing.** To estimate the total uncertainty in the CA precipitation change, we first build a CMIP ensemble including all 128 members from the CMIP5 and CMIP6 models. Then the standard deviation (STD) of the CMIP ensemble is used to estimate the total uncertainty, including the uncertainty arising from internal variability and that from model differences. We note that internal uncertainty might be underestimated in the total uncertainty due to the limited number of ensemble members in each CMIP model[20]. Using all 318 members from the five simulation ensembles to calculate the total uncertainty does not change our results. Consistent with the previous studies[15,17,59], internal uncertainty is determined by the STD of the large ensemble of a given model. Because the different members in a large ensemble of a single model are driven by the same external forcing but differ in their initial conditions, internal variability arising from random climate variations can be estimated by the spread of the ensemble members of a given climate model. Averaging across the three large ensembles yields the multimodel mean internal variability. The contribution of internal variability is calculated as the ratio between the internal variability uncertainty and the total uncertainty, while the intermodel uncertainty can be inferred by the residual assuming the sources of uncertainty are additive[15].

To separate the effect of external forcing in CA precipitation change, the ensemble mean of all the large ensemble members of a single model can be taken as the response to external forcing based on the given model. The variance across such ensemble mean of the three large ensembles represents the model uncertainty in the response to external forcing. To better reduce both the internal variability and intermodel spread in estimating the external frocing, we first average all the members from CMIP5, CMIP6, and the 3 large ensembles, respectively, followed by averaging of the five ensemble mean to get the external forcing effect in Fig. 3.

**Definitions of the IPO, El Niño-like pattern, and indices of key physical processes.** Similar to previous studies[59–61], we define the IPO index as the differences between the SST anomalies averaged within the tropical central-eastern Pacific (10°S to 15°N, 180°E to 90°W) and the North Pacific (25°N to 40°N, 150°E to 140°W). The SST anomalies are obtained by the deviations in each year from the long-term mean for observation and by the deviations of each ensemble member from the ensemble mean for each large ensemble model. The IPO index is defined as the 7-year running average. The spatial patterns and time periods of the IPO simulated in the three large ensembles are reasonable (Supplementary Fig. 2), indicating the reliability of the three models for studying the IPO.

El Niño-like pattern is defined as the SST difference between the tropical central-eastern Pacific (10°S −10°N, 180°E −90°W) and the tropical western Pacific (10°S −10°N, 110°E −140°E). To illustrate the physical mechanisms supporting the IPO's influence on CA precipitation change, several indices are defined based on the regression patterns onto the IPO in Fig. 2, including the precipitation averaged over California (32–42° N, 115° −124° W); the sea level pressure (SLP) averaged over the eastern North Pacific (25–55° N, 180° E-120° W); the westerly jet extension defined by the 200 hPa zonal wind averaged over 20–37° N, 170° E-115° W. We focus on the winter mean (December–January-February), the wet season in CA, in this study.

**Contributions of the IPO to the uncertainty in California precipitation change.** Following ref. [59], we evaluate how much the IPO contribute to the uncertainty in CA precipitation change in several steps: (1) the IPO index is derived for each ensemble member; (2) the CA precipitation variations that are linearly related to the IPO index are removed through a linear regression; (3) the STD of CA precipitation change with and without the IPO can be compared to estimate the uncertainty arising from the IPO; (4) a fixed IPO's influence based on the observed trend of the IPO during 1979–2019 is added to each member after step (2), so that all the members are influenced by the same IPO evolution and can be compared with the observed CA precipitation change during 1979–2019.

## Data availability
The GPCP data is available at https://psl.noaa.gov/data/gridded/data.gpcp.html. The CMAP data is available at https://psl.noaa.gov/data/gridded/data.cmap.html. HadISST is available at https://www.metoffice.gov.uk/hadobs/hadisst/.

The raw outputs of CMIP5 models are available at http://www.ipcc-data.org/sim/gcm_monthly/AR5/Reference-Archive.html. The raw outputs of CMIP6 models are available at available at https://esgf-node.llnl.gov/search/cmip6/. Large ensembles of CanESM2 are available at https://open.canada.ca/data/en/dataset/aa7b6823-fd1e-49ff-a6fb-68076a4a477c. Large ensembles of CESM1 are available at http://www.cesm.ucar.edu/projects/community-projects/LENS/. Large ensembles of MPI-ESM are available at https://esgf-data.dkrz.de/projects/mpi-ge/.

## Code availability
The codes to generate the figures are based on NCAR Command Language (NCL v.6.4.0; https://doi.org/10.5065/D6WD3XH5) and are available at https://zenodo.org/record/5484295#.YTe3LS1h3OQ.

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

## Acknowledgements

This research is supported by the U.S. Department of Energy Office of Science Biological and Environmental Research as part of the Regional and Global Model Analysis and Multisectoral Dynamics program areas. PNNL is operated for the Department of Energy by Battelle Memorial Institute under contract DE-AC05-76RL01830. We acknowledge the World Climate Research Program's Working Group on Coupled Modeling, which is responsible for CMIP5 and CMIP6, and thank the climate modeling groups for producing and making available their model outputs. We also thank the NCAR CESM1 group, MPI-ESM group, and CanESM2 group for making the large ensemble experiments available. For CMIP, the U.S. DOE's Program for Climate Model Diagnostics and Intercomparison provides coordinating support and led the development of software infrastructure in partnership with the Global Organization for Earth System Science Portals.

## Author contributions

L.D. and L.R.L. designed the research. L.D. performed the analysis, drew all the figures, and wrote the first draft of the paper. F.S and J.L. provided comments on the analysis. All authors discussed and commented on the paper.

## Competing interests

The authors declare no competing interests.
