## [Peer Review File · Nature Communications]

REVIEWER COMMENTS

Reviewer #1 (Remarks to the Author):

Summary

In this paper, the authors assess uncertainty in GCM projections of California precipitation in the context of natural variability, specifically the Interdecadal Pacific Oscillation (IPO), elsewhere and most typically referred to as the Pacific Decadal Oscillation (PDO). The paper is exclusively focused on the IPO/PDO as the root cause of uncertainty in GCM projections of precipitation for California and the presentation of results is very detailed. Although I think that the work presented here is of high quality, it is rather narrow in its portrayal/assessment of precipitation projection uncertainty for California and its causes. The broader context is highly relevant, but not presented or even mentioned. Below, I outline a few suggestions for broadening the context and improving the presentation.

Major comment

“Marked uncertainty in California precipitation projections” is the premise, focus and starting point of this paper. This large uncertainty is evident when examining total annual or seasonal winter precipitation (Polade et al. 2014, 2017). But there is also certainty. In fact, on daily timescales, virtually all GCMs agree on the decrease in the frequency of precipitation, particularly in the fall and spring (Pierce et al. 2013, Polade et al. 2014, 2017) and the increase in the heaviest precipitation events (Polade et al. 2017). Moreover, the increase in extreme precipitation is due to atmospheric river (AR) storms, while the decrease of precipitation frequency is due to other (non-AR) storms (Gershunov et al. 2019). Virtually all CMIP5 models agree on these two competing signals. So, there is a lot of certainty, in effect, while the uncertainty comes from the superposition of these two signals – decrease in frequency and increase in intensity. Which of the competing signals dominates is model-specific. This seems to be a first order influence on uncertainty. Natural variability, as assessed in this paper under review, is an important source as well, but probably second-order by comparison.

General comments

IPO/PDO and its impact on CA precipitation

IPO/PDO influence on precipitation was first described in the seminal paper on the PDO – Mantua et al. (1997). Much of that influence is manifested via PDO’s modulation of ENSO teleconnections (Gershunov and Barnett 1998). The PDO does not appear to be a physical mode, but is rather a manifestation of the North Pacific Ocean’s integration of various influences, including ENSO itself (Schneider and Cornuelle 2005, Newman et al. 2016).

ENSO and IPO/PDO in climate models

GCMs are notoriously faulty in their simulation of ENSO. In versions of the CESM model (for example), ENSO is too regular with La Niña almost always following El Niño and less strictly the other way around with possibly a neutral year in between. It would be useful to make sure that the IPO/PDO variability is realistically modeled in the key GCMs used here. Teleconnections, on the other hand, of both ENSO and PDO are realistically represented in most GCMs (Polade et al. 2013).

Specific comments

Lines 30-32. Please provide background and context. Maybe before getting into the uncertainty, it would be useful to mention what we are certain of? Please see major comment above.

Line 32. "...most models project a wetting winter for the region." This is an arguable point. According to Polade et al. 2017, who examined 30 CMIP5 GCMs, there's a pretty even split between the models that project wetter vs drier winters for California. If we look at the entire wet season, more models will show drier conditions as the loss of precipitation occurs preferentially in fall and spring (Pierce et al. 2013).

Lines 33-34. Again, please see and cite Polade et al. (2014, 2017).

While examining internal variability in large ensembles is a worthwhile step towards understanding GCM precipitation projection uncertainty over California, the intro (and perhaps the paper as a whole) is amiss not to consider the basic elements of seasonal precipitation – frequency and intensity of daily precipitation – where we really have certainty. The superposition of these conflicting robust signals together is a major source of uncertainty. See major comment above.

Line 56. "...warming is argued to enhance the probability of severe drought" – please cite Williams et al. (2020).

Lines 56-58. Please describe the correspondence between IPO and PDO and cite the seminal paper – Mantua et al. (1997), which was described the PDO and its relation to precipitation. Would be good to also mention that the PDO influences precipitation over north America via its modulation of ENSO teleconnections (Gershunov and Barnett 1998 and a number of subsequent studies).

Lines 68-70. This statement is inconsistent with results showing less frequent precipitation, particularly in the fall and spring (Pierce et al. 2013, Polade et al. 2014) as well as expansion of subtropical subsidence (e.g. Johanson and Fu 2009, Lau and Kim 2015, etc.), which drives it.

Line 72. It would be useful to clearly define what is meant by "El Niño-like warming".

Line 73. "...the relationship may have been obscured by model shortcoming" – which one?

Line 78. "...La Niña-like warming pattern..." – please define.

Lines 97-99. Do these models simulate IPO/PDO well enough? They certainly have problems with the ENSO cycle. They do simulate ENSO/PDO/IPO teleconnections to precipitation reasonably well (Polade et al. 2013), but for this study, a particularly important question is: do they simulate the interdecadal nature of the IPO reasonably well?

Lines 120-122. The CESM model has a large bias in the ENSO cycle with La Niña always following El Niño. The IPO, being related to ENSO, is unlikely to be realistic either. Some verification results would be useful.

Line 123. In this section again, it would be informative to state the similarity of IPO and PDO and cite key papers, e.g. Mantua et al. (1997).

Line 124. "...the internal climate mode responsible..." – PRIMARILY responsible?

Lines 127-128. "...resembles the positive IPO pattern..." – weather noise is also integrated this way by the north Pacific Ocean, BTW (Schneider and Cornuelle 2005). Do you call it the IPO just because the spatial pattern of correlations somewhat resembles the IPO/PDO? To make sure that it is in fact related to the IPO, rather than that it merely resembles it, please show the time series or, better yet, spectra of the observed and modeled IPO.

Figure 2a shows the strongest correlations in the arctic, which is unlikely to be related to the IPO/PDO. At least a mention of this is in order.

Figure 2b. Mantua et al. (1997) made the connection between PDO and the Aleutian Low. Citation would be appropriate.

Lines 131-133. Really? Connections between IPO/PDO, Aleutian Low, CA precipitation, etc. have not been established? Please see Mantua et al. (1997) and numerous other papers that followed.

In general, a lot has been written about CA precipitation change. The referenced literature here is very limited. It isn't clear, moreover, that CMIP5 GCMs project CA precipitation to increase. See Polade et al. (2014, 2017).

Lines 134-147. See diagnostics of ENSO and PDO/IPO teleconnections to North American precipitation in GCMs (Polade et al. 2013) to check how specific models perform.

Lines 143-147. Please verify that what you call the IPO is indeed interdecadal in GCM simulations.

Line 155. Constraining precipitation uncertainty was also attempted by considering ENSO (Douville et al. 2006, Allen and Luptowitz 2017). Please comment on how these efforts are related.

Lines 191-195. Much of the IPO's impact on CA precipitation comes via its modulation of ENSO teleconnections (Gershunov and Barnett 1998, and subsequent studies). However, ENSO

teleconnections have not been dependable lately regardless of the state of the IPO/PDO.

Lines 246-248. Is this a surprise? Please cite Schneider and Cornuelle (2005).

Lines 260-261. "...CA winter precipitation is influenced by atmospheric circulation and is highly variable." Where is winter precipitation NOT influenced by atmospheric circulation? It is highly variable in CA because extreme storms drive the total precipitation variability (Dettinger et al. 2011).

Line 263. "...this study presents a novel unifying view..." Given what I stated above, I think this is an exaggeration.

Lines 267-268. "...by up to 26%..." – Please provide a figure of the spread of uncertainty reductions by this approach. Between ??% and 26%.....why only provide the upper limit?

Line 282. "...by <30%..." – up to 26%?

Lines 287-288. Is the claim here that the recent drying trend is due to the IPO/PDO? If so, please state and verify this claim.

Lines 290-293. It is suggested that decadal projection uncertainty can be diminished via IPO decadal predictions. It would be useful here to mention the current level of IPO/PDO decadal predictability with the appropriate citations (Meehl et al. 2009, 2014).

References

Atmospheric rivers, floods and the water resources of California.

MD Dettinger, FM Ralph, T Das, PJ Neiman, DR Cayan

Water 3 (2), 445-478, 2011

On the tropical origin of uncertainties in the global land precipitation response to global warming. H

Douville, D Salas-Mélia, S Tyteca

Climate Dynamics 26 (4), 367-385, 2006

Interdecadal modulation of ENSO teleconnections. A Gershunov, TP Barnett

Bulletin of the American Meteorological Society 79 (12), 2715-2726, 1998

Precipitation regime change in Western North America: the role of atmospheric rivers.

A Gershunov, T Shulgina, RES Clemesha, K Guirguis, DW Pierce, ...

Scientific reports 9 (1), 1-11, 2019

A Pacific Interdecadal Climate Oscillation with Impacts on Salmon Production.

NJ Mantua, SR Hare, Y Zhang, JM Wallace, RC Francis

Bulletin of the American Meteorological Society 78 (6), 1069-1080, 1997

Decadal prediction: Can it be skillful?

GA Meehl, L Goddard, J Murphy, RJ Stouffer, G Boer, G Danabasoglu, ...
Bulletin of the American Meteorological Society 90 (10), 1467-1486, 2009

Decadal climate prediction: an update from the trenches

GA Meehl, L Goddard, G Boer, R Burgman, G Branstator, C Cassou, ...
Bulletin of the American Meteorological Society 95 (2), 243-267, 2014

The Pacific decadal oscillation, revisited.

M Newman, MA Alexander, TR Ault, KM Cobb, C Deser, E Di Lorenzo, ...
Journal of Climate 29 (12), 4399-4427, 2016

The key role of heavy precipitation events in climate model disagreements of future annual precipitation changes in California.

DW Pierce, DR Cayan, T Das, EP Maurer, NL Miller, Y Bao, M Kanamitsu, ...
Journal of Climate 26 (16), 5879-5896 2013

Natural climate variability and teleconnections to precipitation over the Pacific-North American region in CMIP3 and CMIP5 models.

SD Polade, A Gershunov, DR Cayan, MD Dettinger, DW Pierce
Geophysical Research Letters 40 (10), 2296-2301, 2013

The key role of dry days in changing regional climate and precipitation regimes.

SD Polade, DW Pierce, DR Cayan, A Gershunov, MD Dettinger
Scientific reports 4 (1), 1-8, 2014

Precipitation in a warming world: Assessing projected hydro-climate changes in California and other Mediterranean climate regions.

SD Polade, A Gershunov, DR Cayan, MD Dettinger, DW Pierce
Scientific reports 7 (1), 1-10, 2017

The forcing of the Pacific decadal oscillation.

N Schneider, BD Cornuelle
Journal of Climate 18 (21), 4355-4373, 2005

Large contribution from anthropogenic warming to an emerging North American megadrought.

AP Williams, ER Cook, JE Smerdon, BI Cook, JT Abatzoglou, K Bolles, ...
Science 368 (6488), 314-318, 2020

Reviewer #2 (Remarks to the Author):

Key Results

An uncertainty quantification of California precipitation due to El Niño-like variability is hypothesized. A 319-member multi-model and large ensemble analysis reveals internal variability contributes >70% and >50% uncertainty in CA precipitation changes and El Niño-like warming known as the Interdecadal Pacific Oscillation (IPO).

Validity

This massive data analysis based on CMIP5 and CMIP6 output provides a recognized approach to decompose uncertainty. The decomposition, however, is based on three single-model large ensembles, CESM1, CanESM2 and MPI-ESM. Each ensemble historic mean SST is regressed to CA precipitation, where the SST pattern indicates the phase of IPO. Results indicate positive IPO correlates with increased CA precipitation. The validity of this result is unclear as correlation coefficients of only one of the large ensembles is provided, CanESM2. This would read better if all three were correlation values were presented CESM1 $r=0.74$, CanESM2 $r=0.79$, MPI-ESM $r=0.82$ for the SLP and IPO trend regression for the 1979-2019 mean. It is important to note that the trends are poor for IPO and CA precipitation for 1979-2019.

Further analysis of the significance of the IPO on CA winter precipitation is based on removing precipitation patterns explained by the IPO index to CA winter precipitation linear regression fit. It should be noted that the IPO to CA DJF precipitation 1979-2019 regressions are 0.34, 0.57 and 0.44 for CESM1, CanESM2 and MPI-ESM, respectively. This consequently yields weak results regarding the findings.

The projected period 2060-2099 is when the scenario uncertainty becomes more dominant than model initial condition internal variability. However, climate system internal variability associated with IPO grows with SST warming during this period. The analysis based on 319 simulation results indicating the CMIP5&6 underestimate such internal variability is valid and of importance for further systematic error reduction.

Significance

Analysis based on large ensemble sets is of importance for delineating sources of uncertainty missed or poorly represented within small ensembles. Combining CMIP5 and CMIP6 along with three subsets of long simulations yields new information on quantifying uncertainty.

Using the large ensemble analysis to constrain CA precipitation uncertainty with the fitted frequency distribution of the CA precipitation trend with the IPO signal removed shows modest reduction in the standard deviation.

Further analysis of the IPO role and Sea Surface Temperature (SST) trend appears to isolate drying signals due to transition shifts in the IPO, implying some improvement toward CA precipitation decadal predictability.

Data and Methodology

Using the CMIP5 and CMIP6 output data is a very large data set that is viewed as the most reliable data for climate analysis. There are bias issues associated with components of these models and this is being addressed by the community and is outside the scope of this study. The statistical methods are adequate and results are reported fairly.

Analytical Approach

Understanding the role of internal variability in the climate system is complex and interconnected. By focusing on one mechanism others may be missed in their inter-related role. The IPO analysis in this study is a massive data crunch that reveals the modest reduction in uncertainty in predicting CA decadal precipitation. The approach is technically sound and provides an understanding of the impact. However, an analysis and connection between persistent blocking high-pressure patterns, IPO trends and SST would strengthen analysis.

Suggested Improvements

This manuscript would benefit if a clear discussion were presented on the teleconnection between the IDO-related jet extension and the presence of persistent high-pressure that steers advected moisture away from CA and its uncertainty. CA's drought occurrence is in conjunction with the high-pressure blocking pattern which directs the storm track north and this is most significant to planners.

Line 122: Indicate the winter months (DJF) of analysis for CA precipitation Lines:
133-140: provide correlation coefficients for all three large ensembles Line 355:
Change NNakamura to Nakamura

Clarity and Context

The manuscript is clear and for the most part the context is adequate.

Reply to Reviewer #1:

We thank the reviewer for the insightful comments and detailed suggestions on how to improve the manuscript. In the following, the original review comments are in *italics* and our responses are in normal font.

Reviewer comments:

In this paper, the authors assess uncertainty in GCM projections of California precipitation in the context of natural variability, specifically the Interdecadal Pacific Oscillation (IPO), elsewhere and most typically referred to as the Pacific Decadal Oscillation (PDO). The paper is exclusively focused on the IPO/PDO as the root cause of uncertainty in GCM projections of precipitation for California and the presentation of results is very detailed. Although I think that the work presented here is of high quality, it is rather narrow in its portrayal/assessment of precipitation projection uncertainty for California and its causes. The broader context is highly relevant, but not presented or even mentioned. Below, I outline a few suggestions for broadening the context and improving the presentation.

Response: We are grateful to the reviewer for the positive comment. In the revision, we have followed your suggestions to broaden the context and improve the presentation. Our point-by-point responses are shown below.

Major comment:

“Marked uncertainty in California precipitation projections” is the premise, focus and starting point of this paper. This large uncertainty is evident when examining total annual or seasonal winter precipitation (Polade et al. 2014, 2017). But there is also certainty. In fact, on daily timescales, virtually all GCMs agree on the decrease in the frequency of precipitation, particularly in the fall and spring (Pierce et al. 2013, Polade et al. 2014, 2017) and the increase in the heaviest precipitation events (Polade et al. 2017). Moreover, the increase in extreme precipitation is due to atmospheric river (AR) storms, while the decrease of precipitation frequency is due to other (non-AR) storms (Gershunov et al. 2019). Virtually all CMIP5 models agree on these two competing signals. So, there is a lot of certainty, in effect, while the uncertainty comes from the superposition

of these two signals – decrease in frequency and increase in intensity. Which of the competing signals dominates is model-specific. This seems to be a first order influence on uncertainty. Natural variability, as assessed in this paper under review, is an important source as well, but probably second-order by comparison.

Response: Thanks for your important comment on the certain and uncertain changes in CA precipitation. We should clarify that in this study, we are interested in CA precipitation uncertainty expressed by the inter-model spread, not just the sign of change. We agree with you that the sign of precipitation frequency and intensity changes are very certain from multi-model ensembles, with decrease in frequency and increase in intensity. As you noted, however, the superposition of these two opposite contributions can result in large spread in model projections of CA precipitation amount, not only for the sign but also for the magnitude. Understanding the sources of this uncertainty provides critical guidance for reducing and communicating uncertainty to stakeholders of the climate information provided by models.

The CA precipitation changes can be decomposed based on many different aspects, such as frequency vs. intensity, thermodynamic vs. dynamical components, extremes vs. non-extremes to shed light on the mechanisms and sources of uncertainty underlying the projected changes. Another fundamental aspect of decomposition is how much of the CA precipitation changes is a response to external forcing vs. a manifestation of internal variability. The different ways of decomposition can also be combined to ask how much of the changes in frequency and intensity is externally forced vs. internal variability. This study emphasizes externally forced vs. internal variability because uncertainty in the response to external forcing is an important and potentially reducible uncertainty factor for targeted future research through model development and observational constraints. Although uncertainty from internal variability is irreducible, given the long timescale of the IPO, improving its decadal prediction may also potentially reduce uncertainty in predicting the decadal trends in CA precipitation in the near future.

Based on your suggestion, we have added discussion of the changes in frequency vs. intensity and the associated certain and uncertain aspects to provide a broader context before motivating our emphasis on uncertainty related to external forcing vs. internal variability in Lines 31-35 of the revised manuscript, and related works (Polade et al. 2014, 2017; Pierce et al. 2013; Gershunov et al. 2019) have been cited. The reason we focus on the uncertainty in response to

externally forced vs. internal variability has been added in Lines 46-51 of the revised manuscript.

General comments:

1) IPO/PDO and its impact on CA precipitation

IPO/PDO influence on precipitation was first described in the seminal paper on the PDO – Mantua et al. (1997). Much of that influence is manifested via PDO’s modulation of ENSO teleconnections (Gershunov and Barnett 1998). The PDO does not appear to be a physical mode, but is rather a manifestation of the North Pacific Ocean’s integration of various influences, including ENSO itself (Schneider and Cornuelle 2005, Newman et al. 2016).

Response: Thanks for your comment. We have added some clarifications on the influence of IPO/PDO on CA precipitation. More specifically, the following statements have been added in the revised manuscript (Lines 66-71): “The Interdecadal Pacific Oscillation (IPO), or Pacific Decadal Oscillation (PDO), is a leading mode of internal variability at the decadal time scale featuring sea surface temperature (SST) variability in the Pacific Ocean. The IPO can be viewed as a manifestation of the integrated influences of the Pacific Ocean, including the El Niño-Southern Oscillation (ENSO) (Schneider and Cornuelle 2005, Newman et al. 2016). As described in previous studies (Mantua et al. 1997; Dai 2013), the IPO impact on CA precipitation is manifested via the interdecadal modulation of ENSO teleconnections (Gershunov and Barnett 1998)”.

2) ENSO and IPO/PDO in climate models

GCMs are notoriously faulty in their simulation of ENSO. In versions of the CESM model (for example), ENSO is too regular with La Niña almost always following El Niño and less strictly the other way around with possibly a neutral year in between. It would be useful to make sure that the IPO/PDO variability is realistically modeled in the key GCMs used here. Teleconnections, on the other hand, of both ENSO and PDO are realistically represented in most GCMs (Polade et al. 2013).

Response: Thanks for your comment. Although ENSO exhibits a quasi-regular biennial behavior in CCSM3 model as you mentioned, the biases have been reduced significantly in CCSM4,

CESM1 and CESM2, which have broader spectral peak with a dominant timescale closer to the observed, and the spatial patterns of the IPO as well as their relationships to ENSO modulations are well simulated in CESM models (Capotondi et al. 2020). We have also examined the IPO simulated by the three main GCMs used in this study (CESM1, CanESM2, MPI-ESM). The results show that the IPO simulated by all the three models show high pattern correlation coefficients (~ 0.7 , statistically significant at the 99% level of confidence) with the observation over the Pacific Ocean. The interdecadal variability of the IPO is also well simulated as indicated by comparing the power spectra between the three models (grey lines) and observation (black lines) shown in Supplementary Fig. 2. In the revision, we have discussed the ENSO biases in CESM, reasonability of the IPO characteristics as well as the IPO teleconnection simulated by the GCMs in Lines 106114. The related paper Polade et al. (2013) and Capotondi et al. (2020) have been cited in the revised manuscript.

Specific comments:

1) Lines 30-32. Please provide background and context. Maybe before getting into the uncertainty, it would be useful to mention what we are certain of? Please see major comment above.

Response: Thanks for your comment. We have added discussion of the certain part in future projections of CA precipitation changes in Lines 31-37 of the revised manuscript as follows: “Despite the very certain sign of precipitation frequency and intensity changes from multi-model ensembles, with decreased frequency and increased intensity, large uncertainty in the changes of annual or winter CA precipitation amount is evident due to the superposition of these two opposite contributions¹⁻⁴. Influenced by both tropical forcing and mid-latitude westerlies⁵, the inter-model spreads of both the signs and magnitudes of CA precipitation changes under warming are large^{2,3,6-13}”.

2) Line 32. “...most models project a wetting winter for the region.” This is an arguable point. According to Polade et al. 2017, who examined 30 CMIP5 GCMs, there’s a pretty even split between the models that project wetter vs drier winters for California. If we look at the entire wet season, more models will show drier conditions as the loss of precipitation occurs preferentially

in fall and spring (Pierce et al. 2013).

Response: Thanks for your comment. We have corrected the statement as “the inter-model spreads of both the signs and magnitudes of CA precipitation changes under warming are large” and cited Polade et al. (2017) and Pierce et al. (2013) here in Lines 35-37 of the revised manuscript. Our emphasis is in the uncertainty of both the sign and magnitude of CA precipitation changes, not only whether models projected drying or wetting (the sign).

3) Lines 33-34. Again, please see and cite Polade et al. (2014, 2017).

While examining internal variability in large ensembles is a worthwhile step towards understanding GCM precipitation projection uncertainty over California, the intro (and perhaps the paper as a whole) is amiss not to consider the basic elements of seasonal precipitation – frequency and intensity of daily precipitation – where we really have certainty. The superposition of these conflicting robust signals together is a major source of uncertainty. See major comment above.

Response: Thanks for your comment. We agree on the importance of providing a larger context on the certainty of CA precipitation projections before we delve into our focus on uncertainty related to external forcing vs. internal variability. Please refer to our response to your Major Comment above. Changes have been made in Lines 31-35 of the revised manuscript.

4) Line 56. “...warming is argued to enhance the probability of severe drought” – please cite Williams et al. (2020).

Response: Williams et al. (2020) has been cited in Line 66 of the revised manuscript.

5) Lines 56-58. Please describe the correspondence between IPO and PDO and cite the seminal paper – Mantua et al. (1997), which was described the PDO and its relation to precipitation. Would be good to also mention that the PDO influences precipitation over north America via its modulation of ENSO teleconnections (Gershunov and Barnett 1998 and a number of subsequent studies).

Response: Thanks for your good suggestion. We have described the correspondence between IPO and PDO, as well as its relationship with CA precipitation via its interdecadal modulation of ENSO teleconnections in Lines 66-71. Mantua et al. (1997) and Gershunov and Barnett (1998) have been cited in the revised manuscript.

6)Lines 68-70. *This statement is inconsistent with results showing less frequent precipitation, particularly in the fall and spring (Pierce et al. 2013, Polade et al. 2014) as well as expansion of subtropical subsidence (e.g. Johanson and Fu 2009, Lau and Kim 2015, etc.), which drives it.*

Response: Thanks for pointing this out. The focus of this study, and that statement in particular, is the winter CA precipitation. We have clarified the CA “winter” precipitation in Line 81 of the revised manuscript. Indeed models projected drying in fall and spring, as noted in the literature cited in your comment as well as our recent studies (e.g., Dong et al. 2019).

7)Line 72. *It would be useful to clearly define what is meant by “El Niño-like warming”.*

Response: Thanks for your good suggestion. We have defined the El Niño-like warming as the stronger SST warming in the tropical eastern Pacific relative to the western Pacific in Line 85 of the revised manuscript.

8)Line 73. *“...the relationship may have been obscured by model shortcoming” – which one?*

Response: Thanks for your comment. Models that can reproduce the observed dynamical teleconnection between the Niño 3.4 SSTs and CA precipitation at inter-annual variability, project more consistent increases in CA precipitation (Allen & Luptowitz, 2017). Model deficiencies that cause uncertainty in this relationship include the CA precipitation sensitivity to Niño 3.4 SSTs, CA precipitation climatology, as well as possible overestimation of tropical convection (Sohn et al. 2016; Allen & Luptowitz, 2017). We have added these discussions in Lines 86-88 of the revised manuscript.

9) Line 78. “...La Niña-like warming pattern...” – please define.

Response: We have defined the La Niña-like warming as the stronger SST warming in the tropical western Pacific relative to the eastern Pacific in Line 90 of the revised manuscript.

0) Lines 97-99. Do these models simulate IPO/PDO well enough? They certainly have problems with the ENSO cycle. They do simulate ENSO/PDO/IPO teleconnections to precipitation reasonably well (Polade et al. 2013), but for this study, a particularly important question is: do they simulate the interdecadal nature of the IPO reasonably well?

Response: We have examined the spatial pattern and power spectra of the IPO simulated by the three models by comparing with the observation based on HadISST. Please refer to our response to General Comment 2 above.

1) Lines 120-122. The CESM model has a large bias in the ENSO cycle with La Niña always following El Niño. The IPO, being related to ENSO, is unlikely to be realistic either. Some verification results would be useful.

Response: Thanks for your comment. The spatial pattern of the IPO simulated by the CESM1 model (Supplementary Fig. 2a) shows high similarity with the observation (Supplementary Fig. 4a), with pattern correlation coefficient of 0.71, statistically significant at the 99% level of confidence. The power spectra of the IPO in observation indicates a cycle of ~13-yr period (black line in Supplementary Fig. 2d), which is well captured by the CESM1 model (grey lines in Supplementary Fig. 2d). Therefore, the CESM1 model has reasonable performance in simulating the main characteristics of the IPO. Note that the large ENSO bias with a quasi-regular biennial behavior found in CCSM3 has been significantly reduced in CCSM4, CESM1 and CESM2, and the spatial patterns of the IPO as well as their relationships to ENSO modulations are well simulated in CESM1 models (Capotondi et al., 2020 and the references therein). The description has been added in Lines 106-113 of the revised manuscript.

2) Line 123. In this section again, it would be informative to state the similarity of IPO and PDO

and cite key papers, e.g. Mantua et al. (1997).

Response: Thanks for your comment. We have stated the similarity of the IPO and PDO and cited Mantua et al. (1997) in Lines 70 and 153 of the revised manuscript.

13) Line 124. “...the internal climate mode responsible...” – *PRIMARILY* responsible? **Response:** We have added “primarily” before “responsible” in Line 149 of the revised manuscript.

14) Lines 127-128. “...resembles the positive IPO pattern...” – weather noise is also integrated this way by the north Pacific Ocean, BTW (Schneider and Cornuelle 2005). Do you call it the IPO just because the spatial pattern of correlations somewhat resembles the IPO/PDO? To make sure that it is in fact related to the IPO, rather than that it merely resembles it, please show the time series or, better yet, spectra of the observed and modeled IPO.

Response: Thanks for your good suggestion. We call it the IPO pattern not only because of its similarity to the IPO spatial pattern, but also because of the reasonably simulated IPO spectra. We have compared the spectra of the observed and modeled IPO in Supplementary Figs. 2d,e,f. The power spectra of the observed IPO features a cycle of ~13-yr period (black line), which is well captured by the three climate models used in this study (grey lines). We have added the statement as “the interdecadal variability of the IPO is also reasonably well simulated as indicated by comparing the power spectra between the three models and observation” in Lines 111-113 of the revised manuscript. Schneider & Cornuelle (2005) has been cited in the revised manuscript.

15) Figure 2a shows the strongest correlations in the arctic, which is unlikely to be related to the IPO/PDO. At least a mention of this is in order.

Response: Thanks for your comment. The highest correlation between the surface temperature and CA precipitation trend occurs north of Bering Sea, which is unlikely to be caused by the IPO, as Ding et al. (2014) found that surface temperature response is absent over there when the observed tropical SST featuring a negative IPO is prescribed in an atmospheric model. We have added this description and cited Ding et al. (2014) in Line 153-156 of the revised manuscript.

16) *Figure 2b. Mantua et al. (1997) made the connection between PDO and the Aleutian Low. Citation would be appropriate.*

Response: Thanks for your comment. We have cited Mantua et al. (1997) in Line 159 of the revised manuscript.

17) *Lines 131-133. Really? Connections between IPO/PDO, Aleutian Low, CA precipitation, etc. have not been established? Please see Mantua et al. (1997) and numerous other papers that followed.*

In general, a lot has been written about CA precipitation change. The referenced literature here is very limited. It isn't clear, moreover, that CMIP5 GCMs project CA precipitation to increase. See Polade et al. (2014, 2017).

Response: We agree that the connections between the IPO/PDO, Aleutian Low, CA precipitation have been well established in previous studies (e.g., Mantua et al. 1997; Dong et al. 2019a). Rather than focusing on the relationship of their time evolution on decadal timescale as done in previous studies, we mainly focus on the effect of the IPO decadal trends on the inter-model spread of the CA winter precipitation trend, which we are able to address due to the recent availability of large ensemble simulations. To avoid misleading statements, we have re-organized the statements in the whole paragraph (Lines 159-180) of the revised manuscript by addressing the uncertainties in the decadal changes. Most CMIP5 models project CA winter precipitation to increase, but decrease in fall and spring (Dong et al., 2019). In this study, we only focus on the CA precipitation in winter, which is the peak season for both CA precipitation and the IPO.

18) *Lines 134-147. See diagnostics of ENSO and PDO/IPO teleconnections to North American precipitation in GCMs (Polade et al. 2013) to check how specific models perform.*

Response: Thanks for your information. Based on Figure 2 of Polade et al. (2013), the three models used in our study perform well in the teleconnection of ENSO/PDO/IPO to North American precipitation. We have mentioned that teleconnections of the IPO can be realistically

represented in most climate models and cited Polade et al. (2013) in Line 113-114 of the revised manuscript.

19) Lines 143-147. Please verify that what you call the IPO is indeed interdecadal in GCM simulations.

Response: We have verified the interdecadal nature of the IPO simulated in the three models (CESM1, CanESM2, MPI-ESM) in Supplementary Fig. 2. Please see details in our response to the General Comment 2, Specific comments 11 and 14.

20) Line 155. Constraining precipitation uncertainty was also attempted by considering ENSO (Douville et al. 2006, Allen and Luptowitz 2017). Please comment on how these efforts are related.

Response: Thanks for your good suggestion. In previous studies, the model ability to capture the ENSO-precipitation relationship at inter-annual timescale is used as a constraint on the uncertainty in precipitation projections. This is based on the model bias perspective as they considered some models have bias in the ENSO teleconnection. Here, we attempt to constrain the CA precipitation decadal change by considering the role of the IPO. Different from emergent constraints that use observations to constrain uncertainty in future projections due to model biases, the state of the IPO can be used to constrain uncertainty in future projections related to internal variability. Related statements have been added in Lines 190-195, and Douville et al. (2006) and Allen and Luptowitz (2017) have been cited in the revised manuscript.

21) Lines 191-195. Much of the IPO's impact on CA precipitation comes via its modulation of ENSO teleconnections (Gershunov and Barnett 1998, and subsequent studies). However, ENSO teleconnections have not been dependable lately regardless of the state of the IPO/PDO.

Response: Thanks for your comment. We have mentioned that the IPO's impact on CA precipitation is manifested via the interdecadal modulation of ENSO teleconnections and cited Gershunov and Barnett (1998) in Lines 70-71 of the revised manuscript. However, as we focus on the decadal trends and long-term changes under warming, not much attention is paid on the ENSO

teleconnection at inter-annual timescale.

22) *Lines 246-248. Is this a surprise? Please cite Schneider and Cornuelle (2005).*

Response: We have cited Schneider and Cornuelle (2005) in Lines 70, 288 and mentioned the connection between the PDO/IPO with ENSO in the revised manuscript.

23) *Lines 260-261. "...CA winter precipitation is influenced by atmospheric circulation and is highly variable." Where is winter precipitation NOT influenced by atmospheric circulation? It is highly variable in CA because extreme storms drive the total precipitation variability (Dettinger et al. 2011).*

Response: We have removed "is influenced by atmospheric circulation" and re-written the sentence as "Located in the path of the Pacific storm tracks and significantly influenced by storms, CA winter precipitation is highly variable" and cited Dettinger et al. (2011) in Lines 301-302 of the revised manuscript.

24) *Line 263. "...this study presents a novel unifying view..." Given what I stated above, I think this is an exaggeration.*

Response: Thanks for your comment. Though the impact of the IPO in CA precipitation on decadal variability has been revealed in previous studies, its effect on the uncertainty of CA winter precipitation change has not been broadly studied or recognized, partly because multi-model large ensemble simulations with perturbed initial conditions are only available in recent years and they are crucial in partitioning uncertainty due to external forcing vs. internal variability. This study provides new information on uncertainty in CA winter precipitation change based on a large ensemble of simulations by combining CMIP5 and CMIP6 along with three sets of large ensemble simulations. We found the importance of internal variability in the uncertainty of CA precipitation decadal trends, future change, and the uncertain El Niño-like warming pattern. The IPO can link the uncertainty in CA precipitation to El Niño-like warming projections mechanistically through its influence on the westerly jet stream in the North Pacific. We also find that removing the IPO's

influence can reduce uncertainty in CA precipitation trends, which is important to the planning for future water security. Further analysis highlights the key role of the IPO phase transition in the observed CA drying trend during the recent decades, implying some improvement toward CA precipitation decadal predictability. To express more accurately, we have revised the statement as “this study links the uncertain CA winter precipitation decadal trends and future changes to the uncertain El Niño-like warming pattern through their respective connections to internal variability” in Line 304-306 of the revised manuscript.

25) Lines 267-268. “...by up to 26%...” – Please provide a figure of the spread of uncertainty reductions by this approach. Between ??% and 26%why only provide the upper limit?

Response: Thanks for your good suggestion. We have added a table showing the uncertainty reduced by removing the IPO’s influence (Table R1). The spread in the CA precipitation trends is reduced after removing the IPO’s influence, with the STD reduced by 0.3%, 16%, 12% for 1979-2019, 26%, 11%, 16% for 2020-2060, and 12%, 25%, 10% for 2061-2099, based on the CESM1, CanESM2, MPI-ESM ensembles, respectively. We have added Table R1 as Supplementary Table 2 and the related description in Lines 198-202 of the revised manuscript.

Table R1 Percentage of the standard deviation of CA precipitation trends reduced by removing the IPO’s influence.

	CESM1	CanESM2	MPI-ESM
1979-2019	0.3%	16%	12%
2020-2060	26%	11%	16%
2061-2099	12%	25%	10%

26) *Line 282. “...by <30%...” – up to 26%?*

Response: As internal variability contributes >70% of uncertainty in the CA precipitation changes under warming, we can infer that model response to external forcing accounts for <30% of the uncertainty in future change. We have clarified how we get the number of <30% by adding “due to >70% of the uncertainty from internal variability” in Lines 321-322 of the revised manuscript.

27) *Lines 287-288. Is the claim here that the recent drying trend is due to the IPO/PDO? If so, please state and verify this claim.*

Response: Yes, the recent drying trend is due to the IPO. We have clarified the statement as “Our findings highlight the importance of internal variability, especially the positive-to-negative phase transition of the IPO, in this observed drying trend” in Lines 335-337 of the revised manuscript. And this conclusion has been verified in Fig. 3d and Supplementary Fig. 10, as well as the related descriptions.

28) *Lines 290-293. It is suggested that decadal projection uncertainty can be diminished via IPO decadal predictions. It would be useful here to mention the current level of IPO/PDO decadal predictability with the appropriate citations (Meehl et al. 2009, 2014).*

Response: Thanks for your good suggestion. We have mentioned the current level of IPO decadal predictability by adding the statement “Near-term predictions of the IPO based on initialized multimodel ensemble decadal hindcasts have shown some promises, with future improvements possible through community activities.” and cited Meehl et al. (2009, 2014) in Lines 342-344 of the revised manuscript.

References

Allen, R. J., & Luptowitz, R. El Niño-like teleconnection increases California precipitation in response to warming. *Nature Communications*, 8, 16055 (2017).

Capotondi, A., Deser, C., Phillips, A. S., Okumura, Y., & Larson, S. M. (2020). ENSO and Pacific

Decadal Variability in the Community Earth System Model Version 2. *Journal of Advances in Modeling Earth Systems*, 12, e2019MS002022. <https://doi.org/10.1029/2019MS002022>. Dettinger, M. D., Ralph, F. M., Das, T., Neiman, P. J., & Cayan, D. R. Atmospheric rivers, floods and the water resources of California. *Water*, 3(2), 445-478 (2011).

Ding, Q., Wallace, J., Battisti, D. et al. Tropical forcing of the recent rapid Arctic warming in northeastern Canada and Greenland. *Nature* 509, 209–212 (2014).

Dong, L., Leung, R., Lu, J. & Song, F. Mechanisms for an Amplified Precipitation Seasonal Cycle in the U.S. West Coast under Global Warming. *J. Climate* 32, 4681–4698 (2019).

Douville, H., Salas-Mélia, D., & Tyteca, S. On the tropical origin of uncertainties in the global land precipitation response to global warming. *Climate Dynamics*, 26 (4), 367-385 (2006)

Gershunov, A., & Barnett, T. P. Interdecadal modulation of ENSO teleconnections. *Bulletin of the American Meteorological Society*, 79 (12), 2715-2726 (1998).

Gershunov, A. et al. Precipitation regime change in Western North America: the role of atmospheric rivers. *Scientific Reports*, 9 (1), 1-11 (2019).

Gershunov, A., Cayan, D. R., Dettinger, M. D., & Pierce, D. W. Natural climate variability and teleconnections to precipitation over the Pacific - North American region in CMIP3 and CMIP5 models. *Geophysical Research Letters*, 40 (10), 2296-2301 (2013).

Mantua, N. J., Hare, S. R., Zhang, Y., Wallace, J. M., & Francis, R. C. A Pacific Interdecadal Climate Oscillation with Impacts on Salmon Production. *Bulletin of the American Meteorological Society*, 78(6), 1069-1080 (1997).

Meehl, G. A. et al. Decadal prediction: Can it be skillful? *Bulletin of the American Meteorological Society*, 90(10), 1467-1486 (2009).

Meehl, G. A. et al. Decadal climate prediction: an update from the trenches. *Bulletin of the American Meteorological Society*, 95 (2), 243-267 (2014).

Newman, M. et al. The Pacific decadal oscillation, revisited. *Journal of Climate*, 29 (12), 4399-4427 (2016).

Pierce, D. W. et al. The key role of heavy precipitation events in climate model disagreements of

future annual precipitation changes in California. *Journal of Climate*, 26 (16), 5879-5896 (2013).

Polade, S. D., Gershunov, A., Cayan, D. R., Dettinger, M. D., & Pierce, D. W. Natural climate variability and teleconnections to precipitation over the Pacific-North American region in CMIP3 and CMIP5 models. *Geophys. Res. Lett.*, 40, 2296–2301 (2013).

Polade, S. D., Pierce, D. W., Cayan, D. R., Gershunov, A., & Dettinger, M. D. The key role of dry days in changing regional climate and precipitation regimes. *Scientific reports*, 4 (1), 1-8 (2014).

Polade, S. D., Gershunov, A., Cayan, D. R., Dettinger, M. D., & Pierce, D. W. Precipitation in a warming world: Assessing projected hydro-climate changes in California and other Mediterranean climate regions. *Scientific reports*, 7 (1), 1-10 (2017).

Schneider, N., & Cornuelle, B. D. The forcing of the Pacific decadal oscillation. *Journal of Climate*, 18(21), 4355-4373 (2005).

Sohn, B.-J., Lee, S., Chung, E.-S. & Song, H.-J. The role of the dry static stability for the recent change in the Pacific Walker circulation. *J. Climate*, 29, 2765–2779 (2016).

Williams, A. P. et al. Large contribution from anthropogenic warming to an emerging North American megadrought. *Science*, 368(6488), 314-318 (2020).

Reply to Reviewer #2:

We thank the reviewer for the insightful comments and detailed suggestions on how to improve the manuscript. In the following, the original review comments are in *italics* and our responses are in normal font.

Key Results:

An uncertainty quantification of California precipitation due to El Nino-like variability is hypothesized. A 319-member multi-model and large ensemble analysis reveals internal variability contributes >70% and >50% uncertainty in CA precipitation changes and El Nino-like warming known as the Interdecadal Pacific Oscillation (IPO).

Validity:

This massive data analysis based on CMIP5 and CMIP6 output provides a recognized approach to decompose uncertainty. The decomposition, however, is based on three single-model large ensembles, CESM1, CanESM2 and MPI-ESM. Each ensemble historic mean SST is regressed to CA precipitation, where the SST pattern indicates the phase of IPO. Results indicate positive IPO correlates with increased CA precipitation. The validity of this result is unclear as correlation coefficients of only one of the large ensembles is provided, CanESM2. This would read better if all three were correlation values were presented CESM1 $r=0.74$, CanESM2 $r=0.79$, MPI-ESM $r=0.82$ for the SLP and IPO trend regression for the 1979-2019 mean. It is important to note that the trends are poor for IPO and CA precipitation for 1979-2019.

Further analysis of the significance of the IPO on CA winter precipitation is based on removing precipitation patterns explained by the IPO index to CA winter precipitation linear regression fit. It should be noted that the IPO to CA DJF precipitation 1979-2019 regressions are 0.34, 0.57 and 0.44 for CESM1, CanESM2 and MPI-ESM, respectively. This consequently yields weak

results regarding the findings.

The projected period 2060-2099 is when the scenario uncertainty becomes more dominant than model initial condition internal variability. However, climate system internal variability associated with IPO grows with SST warming during this period. The analysis based on 319 simulation results indicating the CMIP5&6 underestimate such internal variability is valid and of importance for further systematic error reduction.

Response: We appreciate the positive comments from the reviewer. We want to clarify that all the results we presented in the main text and figures for CanESM2 are also produced for CESM1 and MPI-ESM and provided in the Supplementary, with a brief mention of the Supplementary figures in the main text. For example, we calculated the inter-member regression of SST trends onto the CA precipitation trends among the three large ensembles (CESM1, CanESM2, MPI-ESM) across the past, near-future and far-future decades (1979-2019, 2020-2060, 2061-2099) in Supplementary Fig. 5. All the SST patterns indicate the positive phase of the IPO. Therefore, the IPO decadal trend contributes importantly to the internal variability of the uncertainty in CA winter precipitation trend in the past, near-future, and far-future. Similarly, scatterplots of the SLP trend over the Aleutian low and the IPO trend are also shown for all the three large ensembles across the three periods (Supplementary Fig. 7). The correlation coefficients for 1979-2019 are CESM1 $r=-0.74$, CanESM2 $r=-0.68$, MPI-ESM $r=-0.63$. To reduce clutter in the figures, we only show the results based on CanESM2 during 1979-2019 in the main text and keep the results of other models in the Supplementary. Following your suggestion, we now present the various coefficients in the revised manuscript to provide clear evidence of the consistent results across the three large ensembles (see Lines 174-177).

The inter-member relationships between the IPO and CA winter precipitation are also presented (0.34, 0.57 and 0.44 for CESM1, CanESM2 and MPI-ESM) during 1979-2019, all statistically significant at the 95% confidence level. These correlations for the three large ensembles have been added in Lines 183-186. We agree that removing the IPO's effect from the CA winter precipitation change through linear regression fit has relatively small influence on the uncertainty of CA precipitation trends, as only the variability that is linearly related to the IPO is removed. However, they are non-negligible and indicate the role of IPO in increasing the chance

of both extreme positive and extreme negative precipitation trends. The meaning of the reduced uncertainty by removing the IPO's impact has been described in Lines 202-205. Moreover, two more evidences are provided to illustrate the role of IPO in the CA drying during 1979-2019: (1) Taking the observed IPO transition during 1979-2019 into account, all three large ensembles can well reproduce the observed drying trend in CA precipitation, with magnitudes comparable to the observation (Fig. 3d); (2) Although the ensemble mean of three large ensembles cannot reproduce the observed drying during 1979-2019, some members do reproduce the observed CA drying. In these members, the SST trend pattern also features a negative IPO pattern, similar to the observation (Supplementary Fig. 10). These evidences have been clearly stated in Lines 228-233 and 212-221, respectively.

This study unifies the uncertain CA winter precipitation decadal trends and future changes with the uncertain El Niño-like warming pattern through their respective connections to internal variability. The uncertain CA precipitation changes and El Niño-like warming are physically linked by the IPO, which connects the two by modulating the Aleutian low and westerly jet extension. We have also emphasized these key points in the first paragraph of the concluding remarks of the revised manuscript.

Significance:

Analysis based on large ensemble sets is of importance for delineating sources of uncertainty missed or poorly represented within small ensembles. Combining CMIP5 and CMIP6 along with three subsets of long simulations yields new information on quantifying uncertainty.

Using the large ensemble analysis to constrain CA precipitation uncertainty with the fitted frequency distribution of the CA precipitation trend with the IPO signal removed shows modest reduction in the standard deviation.

Further analysis of the IPO role and Sea Surface Temperature (SST) trend appears to isolate drying signals due to transition shifts in the IPO, implying some improvement toward CA precipitation decadal predictability.

Data and Methodology:

Using the CMIP5 and CMIP6 output data is a very large data set that is viewed as the most reliable data for climate analysis. There are bias issues associated with components of these models and this is being addressed by the community and is outside the scope of this study. The statistical methods are adequate and results are reported fairly.

Analytical Approach:

Understanding the role of internal variability in the climate system is complex and interconnected. By focusing on one mechanism others may be missed in their inter-related role. The IPO analysis in this study is a massive data crunch that reveals the modest reduction in uncertainty in predicting CA decadal precipitation. The approach is technically sound and provides an understanding of the impact. However, an analysis and connection between persistent blocking high-pressure patterns, IPO trends and SST would strengthen analysis.

Suggested Improvements:

This manuscript would benefit if a clear discussion were presented on the teleconnection between the IPO-related jet extension and the presence of persistent high-pressure that steers advected moisture away from CA and its uncertainty. CA's drought occurrence is in conjunction with the high-pressure blocking pattern which directs the storm track north and this is most significant to planners.

Response: Thanks for your good suggestion. Based on Supplementary Fig. 7, linear trends in the IPO show significant relationship with those in the Aleutian low and westerly jet extension. Hence, the teleconnection between the IPO-related jet extension, persistent blocking high-pressure and CA drought can be inferred. To provide further evidence, we calculated the inter-member relationship between the linear trends of the jet extension and the presence of persistent high-pressure directly (Fig. R1). The strong negative relationships exist for all three large ensembles as well as the three periods, with all the correlation coefficients higher than 0.75, statistically

significant at the 99% confidence level. These significant and robust correlations in terms of the uncertainty from internal variability demonstrate the teleconnection between the IPO-related westward retreat of the jet stream and the persistence of high-pressure that steers advected moisture away from CA, contributing to CA drought. In particular, the positive-to-negative phase transition of the IPO contributes to CA drought during 1979-2019. We have added Fig. R1 as Supplementary Fig. 13 of the revised manuscript and the related discussion has been added in Lines 327-338. We also mentioned the significance of these results to planners as “Although uncertainty from internal variability is irreducible, given the long timescale of the IPO, improving its decadal prediction may potentially reduce uncertainty in predicting the decadal trends in CA precipitation in the near future and support stakeholders in planning for changing likelihood of extreme events such as flood and drought” in Lines 338-342 of the revised manuscript.

Fig. R1 Scatterplots of the inter-member relationship between the U200 trend over westerly jet

extension ($\text{m s}^{-1} \text{ 41year}^{-1}$, x-axis) versus SLP trend over Aleutian low (hPa 41year^{-1} , y-axis) in winter during (a-c) 1979-2019, (d-f) 2020-2060, (g-i) 2061-2099 based on 40 members of CESM1 (first column), 50 members of CanESM2 (second column), 100 members of MPI-ESM (third column). Regression lines are shown as red line, and the inter-member correlations (r) are shown at the top-right of each panel.

Line 122: Indicate the winter months (DJF) of analysis for CA precipitation

Response: We have indicated the winter months as December-January-February in Line 30 where it is firstly mentioned in the manuscript.

Lines: 133-140: provide correlation coefficients for all three large ensembles

Response: Thanks for your comment. We have added the correlation coefficients between the trends of Aleutian low and the IPO trends during 1979-2019 for all three large ensembles in Lines 174-177 of the revised manuscript.

Line 355: Change NNakamura to Nakamura

Response: Done.

Clarity and Context:

The manuscript is clear and for the most part the context is adequate.

Response: Thanks for your positive comment.

REVIEWERS' COMMENTS

Reviewer #1 (Remarks to the Author):

The authors have done a thorough job addressing my comments. I find the revised manuscript much improved in the presentation of results and I believe it is ready for publication.

Reviewer #2 (Remarks to the Author):

I have reviewed the two response to reviewer letters and have read through the revised manuscript. The changes that were made have satisfied the needed edits for publication.